# Reduced Apoptotic Injury by Phenothiazine in Ischemic Stroke through the NOX-Akt/PKC Pathway

**DOI:** 10.3390/brainsci9120378

**Published:** 2019-12-15

**Authors:** Yanna Tong, Kenneth B. Elkin, Changya Peng, Jiamei Shen, Fengwu Li, Longfei Guan, Yu Ji, Wenjing Wei, Xiaokun Geng, Yuchuan Ding

**Affiliations:** 1Luhe Institute of Neuroscience, Capital Medical University, Beijing 101100, China; tongyanna@163.com (Y.T.); shenjiameide@163.com (J.S.); fengwulijlu@126.com (F.L.); gp5940@wayne.edu (L.G.); 2Department of Neurology, Luhe Clinical Institute, Capital Medical University, Beijing 101100, China; 3Department of Neurosurgery, Wayne State University School of Medicine, Detroit, MI 48201, USA; kenneth.elkin@med.wayne.edu (K.B.E.); cpeng@med.wayne.edu (C.P.); yding@med.wayne.edu (Y.D.); 4Department of Research & Development Center, John D. Dingell VA Medical Center, Detroit, MI 4820, USA; doc.jiyu@foxmail.com (Y.J.); wwj7168@foxmail.com (W.W.); 5Department of General Surgery, Luhe Clinical Institute, Capital Medical University, Beijing 101100, China; 6China-America Institute of Neuroscience, Xuanwu Clinical Institute, Capital Medical University, Beijing 100053, China

**Keywords:** middle cerebral artery occlusion (MCAO), chlorpromazine and promethazine (C+P), neuroprotection, reperfusion

## Abstract

Phenothiazine treatment has been shown to reduce post-stroke ischemic injury, though the underlying mechanism remains unclear. This study sought to confirm the neuroprotective effects of phenothiazines and to explore the role of the NOX (nicotinamide adenine dinucleotide phosphate oxidase)/Akt/PKC (protein kinase C) pathway in cerebral apoptosis. Sprague-Dawley rats underwent middle cerebral artery occlusion (MCAO) for 2 h and were randomly divided into 3 different cohorts: (1) saline, (2) 8 mg/kg chlorpromazine and promethazine (C+P), and (3) 8 mg/kg C+P as well as apocynin (NOX inhibitor). Brain infarct volumes were examined, and cell death/NOX activity was determined by assays. Western blotting was used to assess protein expression of kinase C-δ (PKC-δ), phosphorylated Akt (p-Akt), Bax, Bcl-XL, and uncleaved/cleaved caspase-3. Both C+P and C+P/NOX inhibitor administration yielded a significant reduction in infarct volumes and cell death, while the C+P/NOX inhibitor did not confer further reduction. In both treatment groups, anti-apoptotic Bcl-XL protein expression generally increased, while pro-apoptotic Bax and caspase-3 proteins generally decreased. PKC protein expression was decreased in both treatment groups, demonstrating a further decrease by C+P/NOX inhibitor at 6 and 24 h of reperfusion. The present study confirms C+P-mediated neuroprotection and suggests that the NOX/Akt/PKC pathway is a potential target for efficacious therapy following ischemic stroke.

## 1. Introduction

Stroke is a costly, prevalent, and debilitating vascular disease [1,2]. Over 7 million Americans ≥ 20 years of age report a past history of stroke, equating to a 2.5% overall prevalence [3,4]. Phenothiazine derivatives chlorpromazine and promethazine (C+P) have been reported to confer neuroprotection against ischemic stroke in many preclinical works [5,6]. It is thought that chlorpromazine protects against neuronal injury by mitigating reactive oxygen species (ROS) accumulation as well as oxidative stress, which results from increased brain metabolism [7]. Promethazine has been observed to reduce mitochondrial permeability during ischemia, mitigating cerebral injury [8,9].

Akt and PKC (protein kinase C), serine/threonine kinases, have been widely investigated for their roles in mediating cell survival, protein synthesis, and metabolism as well as ischemic injury following reperfusion [10]. Akt, found to be a PKC-related protein [11], is heavily involved in the regulation of apoptosis through a variety of downstream pathways [12,13,14]. Much evidence has implicated Akt for its neuroprotective role in cerebral ischemia, finding that Akt expression is associated with neuronal repairment, reduced oxidative stress, and decreased neuronal apoptosis [15,16,17,18]. Additionally, PKC has been studied for its role in ischemic tolerance and reperfusion injury [19,20]. Molecular events during ischemia and reperfusion, including caspase-mediated damage, reduced cerebral blood flow, and increased free radical formation, are at least partially modulated by PKC [21,22,23]. Importantly, one study highlighted that Akt modulates NOX activation and that PKC may act in a pathway upstream from the Akt regulation of NOX [24].

NOX activity may regulate neuroprotection in oxidative stress [25]. Specifically, isoform-specific NOX activity has been reported to cause ROS accumulation, which is implicated in ischemic stroke injury [26]. Considering the Akt/PKC regulation of NOX, the NOX-mediated ROS accumulation, and the C+P-induced ROS reduction, we hypothesize that NOX inhibition given in conjunction with C+P treatment will heighten the neuroprotection offered by C+P treatment alone. We further hypothesize that C+P may achieve neuroprotection through a novel NOX/Akt/PKC pathway. To uncover the underlying mechanism, this study investigated C+P treatment and combined C+P/NOX inhibitor treatment after MCAO-mediated experimental stroke. The expressions of Akt and PKC proteins were tracked, as well as the Bcl family (Bax, Bcl-XL) and downstream effector caspase-3 as biomarkers of cell death. Results from this study may further support neuroprotection resulting from C+P and elucidate a novel and neuroprotective NOX/Akt/PKC pathway.

## 2. Materials and Methods

### 2.1. Subjects

All experimental procedures were approved by the Institutional Animal Investigation Committee of Capital Medical University in accordance with the National Institutes of Health (USA) guidelines for care and use of laboratory animals. Sixty-four adult male Sprague-Dawley rats (280–300 g, Vital River Laboratory Animal Technology Co., Ltd., Beijing, China) were divided prior to surgery into a sham control group that underwent the entire surgical procedure, except for embolization, and three stroke groups with different treatments. The stroke groups underwent MCAO for 2 h at which point they were reperfused. The three groups (n = 8 × 2 each) included: (1) saline (sham treatment), (2) 8 mg/kg C+P given immediately during reperfusion followed by 2.6 mg/kg C+P after 2 h of reperfusion, or (3) 8 mg/kg C+P + NOX inhibitor at the onset of reperfusion followed by 2.6 mg/kg C+P after 2 h of reperfusion. The blood concentration and efficacy of drugs decreased with time, and a single injection of drug was not enough to maintain an optimal blood concentration. In addition, the second dose would be used to enhance the therapeutic effect of the drugs. Thus, we gave a second dose of drug injection to keep blood concentration at an effective level. According to our previous work, a reduced dose (one-third of the original dose) was sufficient to maintain its efficacy. At 24 and 48 h of reperfusion the infarct volume of animals with MCAO were analyzed, and at 6 or 24 h of reperfusion the protein and biochemical measurements were analyzed. The infarcted lesion progressed within 24 to 48 h, and the drugs may slow down infarct progression and provide neuroprotection. We selected an earlier time point (6 h) to study potential molecular changes. This decision was based on previous investigations finding that apoptotic reaction is active during this period.

Neurological deficits were evaluated at 24 h of reperfusion by the modified scoring systems (5 scores) to confirm brain injury after MCAO. The MCAO was regarded as unsuccessful if the scores were less than 2, and the rats of this kind were discarded. There were about 10% of rats with MCAO excluded in our study because of this reason. The mortality rate of each group was less than 10%. In our study, the main cause of death in the ischemic rats was poor operative skills and skull base hemorrhage caused by arterial rupture during insertion of filament, rather than long ischemic time. All data were evaluated with blind analysis.

### 2.2. Focal Cerebral Ischemia

Previously, we described our focal cerebral ischemia model [27,28]. Briefly, the rats were anesthetized with a mixture of 30% oxygen and 70% nitrous oxide in a chamber in addition to 1%–3% isoflurane. Then, they were moved to an operating table and maintained anesthetic with a facemask using 1% isoflurane delivered from a calibrated precision vaporizer. The use of poly-l-lysine-coated intraluminal nylon (4.0) sutures to produce consistent infarcts significantly reduced inter-animal variability. Rats were subjected to a right side MCAO for 2 h by the filaments. Blood pCO_2_ and pO_2_, mean arterial pressure (MAP), and rectal temperature were continuously monitored during the unilateral, two-hour MCAO procedure. Rectal temperatures were kept between 36.5 and 37.5 °C by using a circulating heating pad and a heating lamp.

### 2.3. Phenothiazine Administration

In models with 2 h MCAO followed by reperfusion, a combination of chlorpromazine and promethazine (1:1) was injected intraperitoneally (IP) at the onset of reperfusion at doses of 8 mg/kg in 3 mL saline (as determined by a preliminary study to induce significant neuroprotection). One-third of the original dose was injected 1–2 h later to maintain blood concentration and enhance the effect of the drug. [29]. The drugs were randomized and blinded before administration.

### 2.4. NOX Inhibitor Administration

To determine the potential role of NOX in neuroprotection, apocynin (4-hydroxy-3-methoxy-acetophenone; Sigma, St. Louis, MO, USA) was injected IP with phenothiazine. As mentioned previously, it was dissolved in 90% saline and 10% dimethyl sulfoxide (DMSO) and given at 2.5 mg per kg of body weight [30]. The DMSO content in the final volume solution was only minor (about 0.26%).

### 2.5. Cerebral Infarct Volume

After 2 h of MCAO followed by 24 and 48 h of reperfusion, the resected brains of the rats were cut into slices 2 mm thick (brain matrix) and stained with 2,3,5-triphenyltetrazolium chloride (TTC, Sigma-Aldrich, St. Louis, MO, USA). To minimize error caused by edema, the infarct volume was calculated by an indirect method [28]. The infarct size of the striatum and cortex was also measured and compared at three different levels from anterior +1.00 mm to posterior −4.8 mm to the brain bregma.

### 2.6. Apoptotic Cell Death Assay

Using a photometric enzyme immunoassay, the levels of apoptotic cell death were analyzed via quantification of cytoplasmic histone-associated DNA fragments [31] (Cell Death Detection ELISA; Roche Diagnostics, Indianapolis, IN, USA). We followed the manufacturer’s protocol and made slight modifications as described previously [32]. In brief, frozen tissue (10 mg) of rats that contained the frontoparietal cortex and striatum was finely cut and incubated on a shaker for 20 min at room temperature in 0.1 M citric acid solution containing 0.5% Tween-20. The solution was then centrifuged at 2000 rpm to precipitate the nuclei. Pellet was discarded, and supernatant (20 µL) was retained as sample. To estimate the concentration of cells in the sample solution, a BCA protein assay was used on the final sample solution after dilution to 0.1 µg/µL with incubation buffer. With a multimode detector (Beckman DTX-880, Beckman Coulter, Inc., Wals, Austria), absorbance of 405 nm was detected.

### 2.7. NOX Activity Assay

Brain samples of the ipsilateral MCAO supplied regions were homogenized in buffer (120 mM NaCl, 4.8 mM KCl, 1.2 mM MgSO_4_, 2.2 mM CaCl_2_, 0.15 mM Na_2_HPO_4_ at pH 7.4, 0.4 mM KH_2_PO_4_, 20 mM HEPES, 5 mM NaHCO_3_, and 5.5 mM glucose), containing a mixture of phenylmethylsulphonyl fluoride and a protease inhibitor (Thermo Fisher Scientific, Waltham, MA, USA). Homogenate (20 μL) with 80 μL homogenizing buffer supplemented with 6.25 μM lucigenin was then added to a 96-well luminescence plate. The reaction was initiated by the addition of NADPH (100 μM), and luminescence was recorded with a DTX-880 multimode every 30 s detector for 5 min [33].

### 2.8. Western Blotting

Protein levels of PKC-δ, p-Akt, Caspase-3, Bax, and Bcl-XL were detected by western blotting. Brain tissues containing the frontoparietal cortex and dorsolateral striatum were processed as described previously [34] and incubated with primary antibodies (polyclonal rabbit anti-PKC-δ at 1:5000, Santa Cruz Biotechnology, Santa Cruz, CA, USA; polyclonal rabbit anti-phospho-Akt 1:1000, Cell Signaling Technology, Danvers, MA, USA; rabbit polyclonal anti-caspase-3 antibody, 1:1000, Santa Cruz Biotechnology, Santa Cruz, CA, USA; rabbit polyclonal anti-cleaved caspase-3 antibody, 1:5000, Santa Cruz Biotechnology, Santa Cruz, CA, USA; rabbit polyclonal anti-BAX antibody, 1:500, Santa Cruz Biotechnology, Santa Cruz, CA, USA; mouse monoclonal anti-Bcl-XL, 1:200, Santa Cruz Biotechnology, Santa Cruz, CA, USA) at 4 °C. Western blot images were analyzed by an image analysis program (ImageJ 1.42, National Institutes of Health, Bethesda, MD, USA) to quantify protein expression in terms of relative image density.

### 2.9. Statistical Analysis (SPSS Software, Version 19, SPSS Inc.)

Sample size of 8 in each group was determined by power analysis based on previous study [29]. A power analysis was conducted for the experiment based on preliminary data. Given the large mean difference and small standard deviation (SD) in previous studies [29], we predicted that the effective size for our proposed study was about 1.00 or above, suggesting a small sample size. Thus, to attain a power exceeding 95% (*p* = 0.05, power = 0.95) and yield statistically significant results (*p* < 0.05) using ANOVAs, we proposed a sample size of eight animals for each group. Study data were described as mean ± standard error (SE). Differences among groups were assessed using one-way analysis of variance or Student’s *t* test with a significance level of *p* < 0.05. Post hoc comparison between groups was achieved using the least significant difference (LSD) method.

## 3. Results

### 3.1. Physiological Parameters.

There were no significant differences in blood MAP, pO_2_, or pCO_2_ between the groups (Table 1).

### 3.2. Cerebral Infarct Volume

Ischemic injury produced a substantial infarct volume (50.0% ± 2.5%) at 24 h of reperfusion. A significant reduction of infarct volume was induced by C+P (33.7% ± 6.0%) and C+P/NOX inhibitor (28.5% ± 3.0%) (Figure 1A,B). However, no significant difference in infarct volume was found between the two treatment groups.

To further determine the progression on infarction at a later time point and the effect of DMSO, additional experiments were conducted to show the infarct volume at 48 h of reperfusion (Figure 1C,D). Similarly, as compared to the no treatment group (39.1% ± 3.1%), a significant reduction of infarct volume was induced by C+P (23.1% ± 5.5%). No significant difference was found between the no treatment group (39.1% ± 3.1%) and the DMSO group (37.1 ± 4.4%).

### 3.3. Cell Death

As compared to the sham-operated group (reference as 1, not shown), the stroke group exhibited increased apoptotic cell death (*p* < 0.01). C+P treatment significantly decreased cell death at both 6 and 24 h post-ischemia (*p* < 0.05) (Figure 2). C+P/NOX-inhibitor treatment also resulted in a significant decline in cell death at both time points (*p* < 0.01), though no further difference was found as compared to C+P monotherapy.

### 3.4. NOX Activity

Ischemia resulted in significantly increased NOX activity at 6 and 24 h of reperfusion as compared to the shame control (reference as 1, not shown) (Figure 3). As compared to the saline treatment, NOX activity was significantly reduced at 6 and 24 h post-ischemia by both C+P treatment and C+P/NOX inhibitor treatment. Again, no significant difference was found between the two treatment cohorts. 

### 3.5. PKC-δ and p-Akt Protein

PKC-δ protein expression was increased in the stroke group at both 6 and 24 h of reperfusion as compared to the shame control (reference as 1) (*p* < 0.01) (Figure 4A). C+P treatment significantly reduced PKC-δ protein expression at both 6 h (*p* < 0.01) and 24 h (*p* < 0.05). C+P/NOX inhibitor treatment precipitated a further significant decrease at both 6 h (*p* < 0.01) and 24 h (*p* < 0.01) time points (Figure 4A).

Compared to the sham control (referenced as 1), p-Akt levels were significantly decreased (*p* < 0.01) after 2 h MCAO followed by 6 and 24 h of reperfusion (Figure 4B). p-Akt protein expression was significantly increased in both treatment groups at 6 (*p* < 0.01) and 24 h (*p* < 0.01) post-ischemia.

### 3.6. Caspase-3 Activity and Bax Protein

Stroke increased cleaved and uncleaved caspase-3 protein at 6 and 24 h of reperfusion when compared to sham control (*p* < 0.01) (Figure 5A,B). Cleaved caspase-3 protein was significantly reduced by C+P and C+P/NOX inhibitors at both 6 and 24 h of reperfusion (*p* < 0.01) (Figure 5A), while uncleaved caspase-3 protein was only reduced by C+P/NOX inhibitor at 6 h post-ischemia (*p* < 0.05) (Figure 5B). At 24 h post-ischemia, caspase-3 levels were significantly reduced in both C+P and C+P/NOX inhibitor (*p* < 0.01). Stroke increased Bax protein at both 6 and 24 h of reperfusion when compared to sham control (*p* < 0.01) (Figure 5C). Bax protein was significantly reduced by C+P and C+P/NOX inhibitor at both time points, while no significant difference was found between the two treatment groups.

### 3.7. Bcl-XL Protein

There was a decrease (*p* < 0.01) in protein levels of Bcl-XL (Figure 5D) after stroke with 6 and 24 h of reperfusion. At 6 and 24 h post-ischemia, both C+P and C+P/NOX-inhibitor treatments showed a significant increase in Bcl-XL protein expression. There was no significant difference in Bcl-XL protein expression between the two treatment groups.

## 4. Discussion

The present study provides further evidence that post-ischemic C+P treatment is neuroprotective. C+P have been commonly used in combination for decades. Many previous studies [35,36,37] in ischemia have demonstrated that combined chlorpromazine and promethazine could induce “artificial hibernation” and neuroprotection. While using them alone, their antipsychotic and sedative effects are more obvious. Since the aim of the current study was to explore the neuroprotective mechanism of phenothiazines, we chose combined chlorpromazine and promethazine rather than individual drug. In the study, the intervention drugs were dissolved in DMSO. The concentration of DMSO in this study was extremely low (less than 0.26%) and did not induce a beneficial effect. In addition, temperature was not controlled during the procedure of phenothiazine administration in this study. Our previous studies have found that temperature is not a primary determinant of neuroprotection [29]. The neuroprotection offered by C+P and C+P/NOX inhibitor administration is demonstrated by the significant reductions in cell death and infarct volume, as well as reductions in Bax and caspase-3, and increases in Bcl-XL. Between treatment groups, the comparison of reductions in NOX activity, PKC expression, infarct volume, and cell death in addition to the comparison of increase in p-Akt expression indicates that NOX inhibition may not confer additional neuroprotection when administered in conjunction with C+P. These results suggest that C+P confers its therapeutic effects via NOX inhibition. The present study further suggests that an integrated NOX/Akt/PKC pathway is involved in augmenting or precipitating neuroprotection following ischemic stroke (Figure 6).

NOX inhibition has been previously recognized as a potential innovation in the prevention of neurodegeneration following acute ischemic stroke; isoform-specific NOX inhibitors have further been identified as potential, clinically effective neuroprotectants through reduction in oxidative stress [26]. Previous studies have also demonstrated that C+P-induced neuroprotection results in decreased ROS production and stabilized brain metabolism, which is associated with p-Akt and PKC protein expression [29]. The lack of additional neuroprotection by NOX inhibitors suggests that C+P-induced neuroprotection is achieved, at least partially, through NOX activity.

In previous studies, neuroprotection has been associated with reduced PKC, and increased Akt, expression [29] that is likely achieved through mitigation of oxidative stress resulting from ROS generation and hyperglycemia, which are well-known effects of ischemic injury [38]. PKC is upregulated during ischemic stroke concurrent with accumulation of ROS, which may indicate various glucose metabolism mechanisms involving Akt/PKC [39,40,41,42,43]. Akt has been identified as a potential target for stroke therapy through possible modulation of insulin and phosphoinositide 3-kinases (PI 3-kinases), which may stabilize normo-metabolism and ROS recovery processes [44,45]. Additionally, it was demonstrated that metabolic dysfunctions in ischemic stroke are stabilized with C+P treatment by depressing brain metabolism and ROS accumulation [46,47]. Taken together, it is reasonable to implicate the Akt/PKC pathway as a means of neuroprotection following C+P administration.

The therapeutic potential of the NOX/Akt/PKC pathway was also indicated by variations in Bcl-XL, Bax, and their associated caspases, specifically caspase-3. Bcl-XL and Bax are known to regulate apoptosis and offer protection from neuronal injury following cerebral ischemia through self-regulation and regulation of caspases, mediators of apoptotic cell death [48,49,50,51,52]. This neuroprotection was reported to be associated with increased levels of anti-apoptotic Bcl-XL and reduced levels of pro-apoptotic Bax and caspase-3 [53]. In the present study, NOX/Akt/PKC pathway inhibition, which led to an increase in Bcl-XL and a decrease in Bax and caspase-3, indicates that it is involved in neuronal apoptotic mechanisms and, therefore, has potential for future stroke therapy.

## 5. Conclusions

The present study supports the efficacy of C+P-mediated neuroprotection following acute ischemic stroke and suggests that this neuroprotection is achieved through an integrated NOX/Akt/PKC pathway. Future research should focus on evaluating C+P administration as a potential means of precipitating the neuroprotective effects of NOX inhibition as well as elucidating the role of NOX in the NOX/Akt/PKC pathway.

## Figures and Tables

**Figure 1 brainsci-09-00378-f001:**
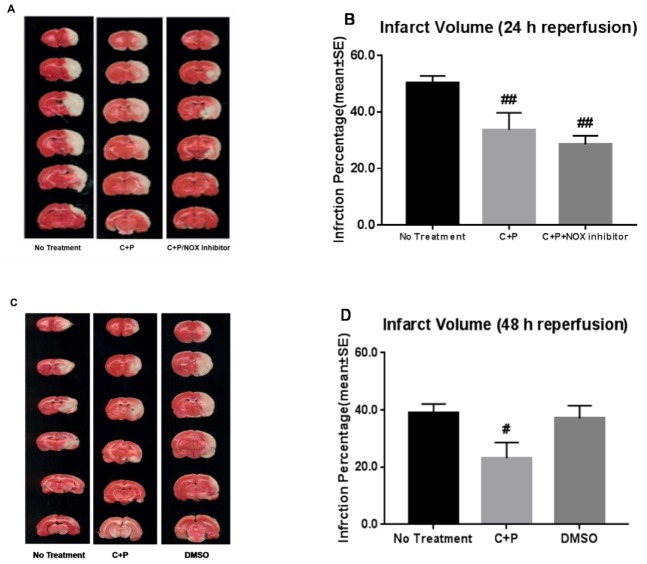
Infarct volume reduction by control treatment, chlorpromazine and promethazine (C+P) treatment, and C+P/NADPH oxidase (NOX) inhibitor treatment. 2,3,5-Triphenyltetrazolium chloride (TTC) histology depicts (**A**) the cortex and striatum at three different levels supplied by the middle cerebral artery (MCA) from anterior +1.00 mm to posterior −4.8 mm to the bregma at 24 h of reperfusion. (**B**) Percentage of infarct volume reduction (mean ± standard error (SE)) with no treatment (50.0% ± 2.5%), C+P treatment (33.7% ± 6.0%), and C+P/NOX inhibitor treatment (28.5% ± 3.0%) at 24 h of reperfusion. While no significant difference in infarct volume was produced between cohorts, there was a significant reduction in both cohorts when compared to no treatment (^##^
*p* < 0.01). In addition, at 48 h of reperfusion (**C**,**D**), infarct volume in ischemic rats (39.1% ± 3.1%) was significantly reduced by C+P treatment (23.1% ± 5.5%) (^#^
*p* < 0.05), while DMSO alone did not induce any neuroprotection (37.1% ± 4.4%). MCA, middle cerebral artery, C+P, chlorpromazine and promethazine.

**Figure 2 brainsci-09-00378-f002:**
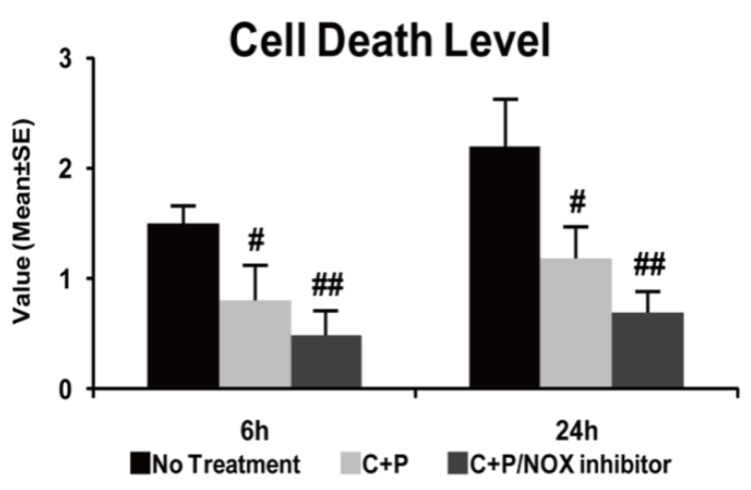
Apoptotic cell death photometric enzyme immunoassay in control treatment, C+P treatment, and C+P/NOX inhibitor treatment. ELISA quantified the degree of apoptosis via 405 nm wavelength absorbance. C+P treatment significantly reduced cell death (mean ± SE) at 6 and 24 h, and C+P/NOX inhibitor treatment augmented the reduction in cell death at each time point. Cell death level at 6 h: no treatment 1.5 ± 0.2, C+P 0.8 ± 0.3, C+P/NOX inhibitor 0.5 ± 0.2; cell death level at 24 h: no treatment 2.2 ± 0.4, C+P 1.2 ± 0.3, C+P/NOX inhibitor 0.7 ± 0.2 (^#^
*p* < 0.05, ^##^
*p* < 0.01).

**Figure 3 brainsci-09-00378-f003:**
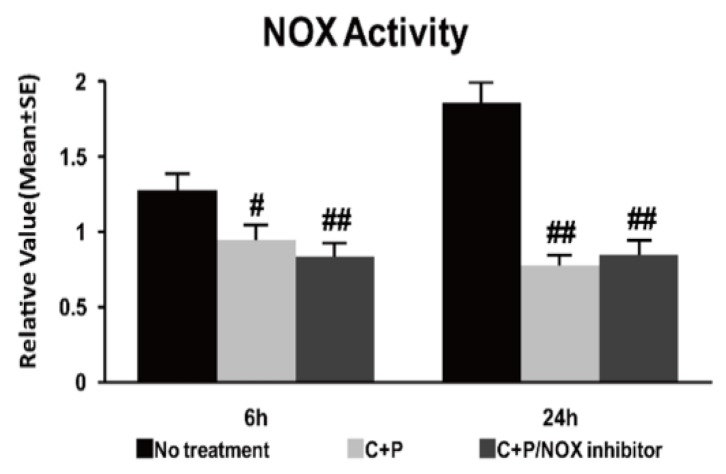
NOX activity luminescence assay in control treatment, C+P treatment, and C+P/NOX inhibitor treatment cohorts. C+P treatment and C+P/NOX inhibitor treatment both produced decreased NOX activity (mean ± SE) at both 6 and 24 h, though there was no significant difference between treatment cohorts at 6 or 24 h of reperfusion. NOX activity at 6 h: no treatment 1.3 ± 0.1, C+P 1.0 ± 0.1, C+P/NOX inhibitor 0.8 ± 0.1; NOX activity at 24 h: no treatment 1.9 ± 0.1, C+P 0.8 ± 0.1, C+P/NOX inhibitor 0.8 ± 0.1 (^#^
*p* < 0.05, ^##^
*p* < 0.01).

**Figure 4 brainsci-09-00378-f004:**
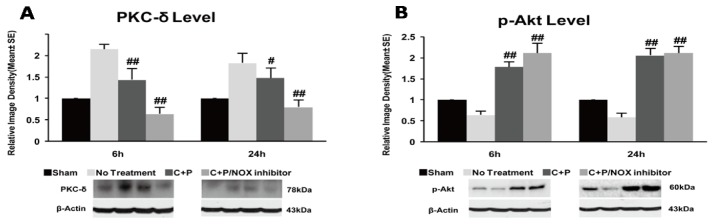
PKC (protein kinase C) and p-Akt protein expression and western blotting in control treatment, C+P treatment, and C+P/NOX inhibitor treatment cohorts. Brain tissue containing the dorsolateral striatum and the frontoparietal cortex were processed and used to determine protein expression (mean ± SE). (**A**) PKC protein expression significantly decreased in each treatment cohort at 6 and 24 h. PKC protein expression was further decreased in the C+P/NOX inhibitor treatment group at 6 and 24 h. PKC level at 6 h: no treatment 2.2 ± 0.1, C+P 1.4 ± 0.3, C+P/NOX inhibitor 0.6 ± 0.2; PKC level at 24 h: no treatment 1.8 ± 0.2, C+P 1.5 ± 0.2, C+P/NOX inhibitor 0.8 ± 0.2. (**B**) p-Akt protein expression was elevated in both treatment cohorts at 6 h and 24 h. p-Akt level at 6 h: no treatment 0.6 ± 0.1, C+P 1.8 ± 0.1, C+P/NOX inhibitor 2.1 ± 0.2; p-Akt level at 24 h: no treatment 0.6 ± 0.1, C+P 2.1 ± 0.2, C+P/NOX inhibitor 2.2 ± 0.1 (^#^
*p* < 0.05, ^##^
*p* < 0.01). Original Western blot images are provided in Appendix A.

**Figure 5 brainsci-09-00378-f005:**
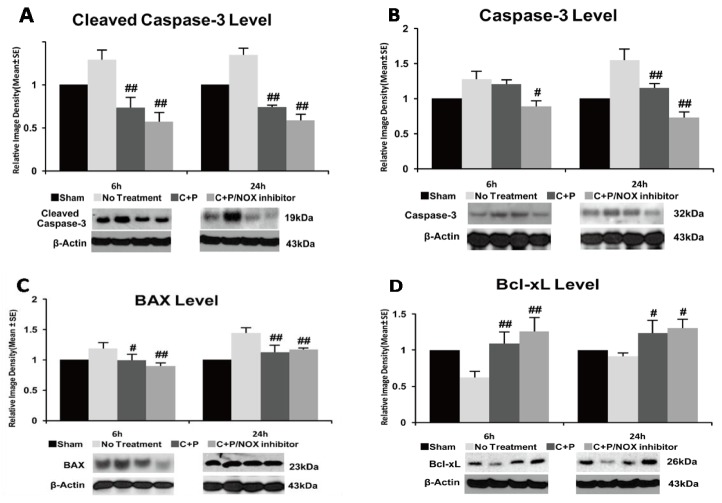
Cleaved and uncleaved caspase-3, Bax, and Bcl-XL protein expression, and Western blotting in control treatment, C+P treatment, and C+P/NOX inhibitor treatment. Brain tissue containing the dorsolateral striatum and frontoparietal cortex were processed and used to detect protein levels (mean ± SE). (**A**) Cleaved caspase-3 protein was significantly reduced by both C+P and C+P/NOX inhibitor treatment cohorts at 6 and 24 h. Cleaved caspase-3 level at 6 h: no treatment 1.3 ± 0.1, C+P 0.7 ± 0.1, C+P/NOX inhibitor 0.6 ± 0.1; cleaved caspase-3 level at 24 h: no treatment 1.3 ± 0.1, C+P 0.7 ± 0.0, C+P/NOX inhibitor 0.6 ± 0.1. (**B**) Uncleaved caspase-3 protein was significantly reduced only in the C+P/NOX inhibitor cohort at 6 h. At 24 h, both C+P and C+P/NOX inhibitor cohorts resulted in decreased caspase-3; caspase-3 protein expression in the C+P/NOX inhibitor cohort was further decreased in comparison to the C+P cohort. Caspase-3 level at 6 h: no treatment 1.3 ± 0.1, C+P 1.2 ± 0.1, C+P/NOX inhibitor 0.9 ± 0.0; PKC level at 24 h: no treatment 1.5 ± 0.2, C+P 1.2 ± 0.1, C+P/NOX inhibitor 0.7 ± 0.1. (**C**) At 6 h, Bax protein expression was reduced in both C+P and C+P/NOX inhibitor treatment cohorts. At 24 h, both C+P and C+P/NOX inhibitor treatment cohorts exhibited a significant decrease in Bax protein expression. Bax level at 6 h: no treatment 1.2 ± 0.1, C+P 1.0 ± 0.1, C+P/NOX inhibitor 0.9 ± 0.0; Bax level at 24 h: no treatment 1.4 ± 0.1, C+P 1.1 ± 0.1, C+P/NOX inhibitor 1.2 ± 0.0. (**D**) At 6 h, both C+P and C+P/NOX inhibitor treatment cohorts resulted in a significant increase in Bcl-XL protein expression. At 24 h, both C+P and C+P/NOX inhibitor treatment groups also produced a significant increase. Bcl-XL level at 6 h: no treatment 0.6 ± 0.1, C+P 1.1 ± 0.2, C+P/NOX inhibitor 1.3 ± 0.1; Bcl-XL level at 24 h: no treatment 0.9 ± 0.0, C+P 1.2 ± 0.2, C+P/NOX inhibitor 1.3 ± 0.1 (^#^
*p* < 0.05, ^##^
*p* < 0.01).

**Figure 6 brainsci-09-00378-f006:**
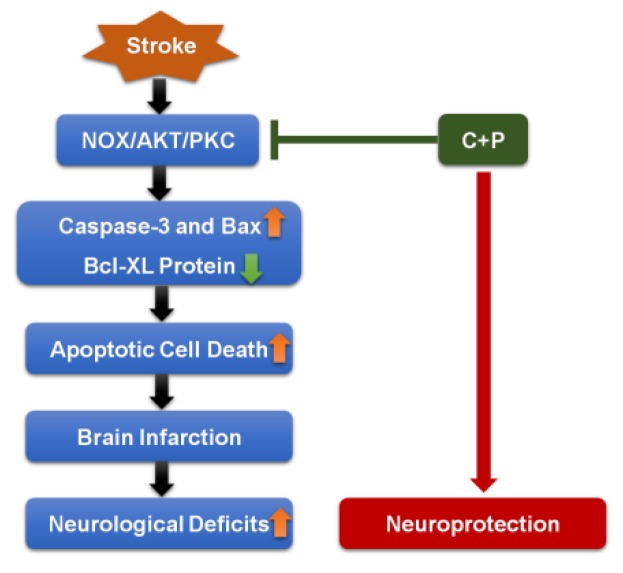
Pathway of neuroprotection from phenothiazines. C+P suppresses expression of Caspase-3 and Bax, while enhancing Bcl-XL expression via inhibition of the NOX-Akt/PKC pathway. This regulation results in decreased apoptotic cell death, leading to a subsequent reduction in brain infarction and a mitigation of neurological deficits.

**Table 1 brainsci-09-00378-t001:** Physiological parameters during surgery.

	Sham Control	No Treatment	C+P	C+P/NOX Inhibitor
**MAP (mmHg)**				
Pre MCAO	86.6 ± 2.5	85.8 ± 2.5	86.5 ± 2.9	84.2 ± 2.1
Onset of reperfusion	88.6 ± 2.8	87.9 ± 2.8	86.3 ± 3.6	89.1 ± 3.3
After reperfusion	83.2 ± 3.0	86.5 ± 2.8	84.5 ± 4.4	86.2 ± 3.7
**pO_2_ (mmHg)**				
Pre MCAO	134.8 ± 5.6	135.5 ± 5.0	133.6 ± 5.3	134.9 ± 5.7
Onset of reperfusion	131.3 ± 6.7	132.7 ± 5.8	133.1 ± 6.3	131.5 ± 6.7
After reperfusion	134.1 ± 8.8	140.1 ± 7.3	136.9 ± 6.2	139.1 ± 7.7
**pCO_2_ (mmHg)**				
Pre MCAO	45.0 ± 1.4	46.2 ± 1.9	46.8 ± 2.9	45.5 ± 1.6
Onset of reperfusion	43.6 ± 3.2	44.4 ± 2.5	43.2 ± 2.7	43.9 ± 2.7
After reperfusion	44.6 ± 5.0	45.6 ± 2.7	46.7 ± 3.3	48.1 ± 4.3

Sham Control indicates that the control group underwent the entire surgical procedure except for embolization; MAP, mean arterial pressure; MCAO, middle cerebral artery occlusion, C+P, chlorpromazine and promethazine.

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
