# Peer review of "Reduced Apoptotic Injury by Phenothiazine in Ischemic Stroke through the NOX-Akt/PKC Pathway"

_brainsci, 2019, doi:10.3390/brainsci9120378_

Round 1

Reviewer 1 Report

Tong and colleagues have edited the manuscript appropriately. The only minor thing I would encourage them to move is the comment on ipsilateral tissue samples, which they expanded in 2.1 but I encourage them to move it to the section on the specific assays where they still just say 'brain samples'. Otherwise the manuscript is now appropriate for publication.

Reviewer 2 Report

Tong et al. report that administration of phenothiazines is neuroprotective in a rat model of ischemic stroke. The mechanism they have identified is reduced apoptosis through reduced NOX activity, PKC expression and increased phospho-Akt, although the exact interactions are not clearly delineated. While I think the underlying observations are of interest, I have some suggestions to the authors.

The introduction should be clearer that the neuroprotective effects of phenothiazines are based on preclinical work. The statement currently follows a brief description of stroke epidemiology (lines 36-39), which could be inferred as describing clinical efficacy. Further detail should be included on the exact MCAO methodology (lines 95-104) beyond referring to their previous work. In particular, which vessel was used to introduce the filament? Was the CCA recanalized as well after two hours? It is excellent that the authors recorded blood gas values and MAP; I would encourage reporting these figures in the results, to exclude any baseline differences between groups. At what time point was behavioral data collected (lines 88-91)? Functional improvements are a more informative measure of neuroprotection than histological or biochemical changes, and it would be useful to known if there was any difference between the groups. The cell death assay and Western blotting was performed using frontoparietal cortex and striatum (lines 126-127). How did the authors ensure that the same regions were sampled in each case, particularly as infarct volumes differed between groups – was the definition based on absolute coordinates (e.g. relative to bregma)? I.e. is it not possible that the differences between groups were simply a reflection of one group including more viable tissue, whereas the other group had more infarcted tissue in the sample? On line 268, the authors state that temperature was not controlled in this study (and refer to a previous paper suggesting it temperature is not an important factor), while the methods report keeping rats between 36.5 and 37.5 degrees. It is extremely important to clarify this, given the potent neuroprotective effects of hypothermia, which has been known to confound previous studies of neuroprotectants (e.g. the NMDA antagonists). The authors repeatedly state that Akt protein expression was increased, but the data presented only show phosphorylated Akt. Was total Akt ever quantified? Is this a change in expression or only phosphorylation? I would encourage the authors to include the full Western blot images as a supplementary, as it is very difficult to inspect the blots from the current images. The presentation would be enhanced by an illustration showing the signaling pathways that the authors have identified as putative targets of phenothiazines.

Round 2

Reviewer 2 Report

I thank the authors for addressing most of the clarifications I asked for, in particular the methodological queries about the MCAO model. The only further suggestion I have is to upload the raw Western blot images - currently the supplementary materials include cropped bands only. My original comments on this were not phrased clearly enough, but what I was referring to were full, uncropped images showing all lanes and molecular weight markers.

Author Response

Please find the part of our raw data for the present study.
However, we DO NOT wish to publish the very raw data as the supplements.

This manuscript is a resubmission of an earlier submission. The following is a list of the peer review reports and author responses from that submission.

Round 1

Reviewer 1 Report

The authors aimed to investigate the neuroprotective mechanisms of phenothiazine following experimental stroke, specifically through the NOX-Akt/PKC pathway. This manuscript is potentially important to stroke research. However, there are some concerns and here are some suggestions to improve the study/manuscript. 

There is a lot of key information relating to the animal experiments which has been omitted - eg power calculation to determine group size, details on numbers used for each experiment western blotting, cell death assay etc., inclusion/exclusion criteria. Figure 4 and 5, please include blots for sham control.  It would give more credence to the results if the functional outcome of the neuroprotective effects were also assessed eg improvement in neurological deficits.  The authors only investigated infarct size at 24 hr post reperfusion. Perhaps a later time point at 7 days is more appropriate to recapitulate the neuroprotective effects.  It is difficult to specifically to make conclusion whether the protective effects are from chlorpromazine or promethazine when both compound were administered together. 

Minor:

Shame group to sham group Page 2 line 45 - please explain what is neuronal rehabilitation 

Reviewer 2 Report

Tong and colleagues explore the potential for phenothiazines chlorpromazine and promethazine to be used as post-stroke therapy to reduce cell death. Overall the paper is well written and well presented and the conclusions are sound. Some minor points which would benefit from clarification are as follows:

The authors say that a reduced dose of the drug is used later to 'enhance effects' but the reasons behind this are not clear. The authors reference their own work from 2017 but in this paper the reasons behind this second dose are not scientifically clear. In their 2017 paper, the authors demonstrated that there was a significant reduction in core body temperature using 8mg C+P, hypothermia is neuroprotective and when they control for temperature in the original study they reduce the efficacy. Was temperature controlled for after the i.p. injections in this study? The drugs are dissolved in DMSO yet there is no DMSO control group, the control group appears to be saline, could the authors comment on this? The use of 'brain samples' for some of the biochemistry assays needs to be clarified. Did the authors take only ipsilateral tissue or both ipsilateral and contralateral? The use of 6 hours and 24 hours is not explained. 24 hours seems to be standard for investigating infarct and I appreciate that many of the processes are going to be initiated at 6 hours but this is not clear from the manuscript. The representative blots in figure 4A are unconvincing when looked at in conjunction with the graphs. The large drop between no treatment and C+P is just not obvious from the blot shown. Phenothiazines are known to have effects on blood flow, as cerebral blood flow is important for post-stroke recovery, have the authors measured this in their studies? The authors say they analysed all data blind but were drugs randomized and blinded before administration?

Overall the manuscript needs some minor editing for English but reads well.

Round 2

Reviewer 1 Report

Page 4 line 153, Sample size of 8 in each group was determined by power analysis based on previous study. Please provide the reference of the previous study. 

The b-actin bands in Fig 4A are identical to 4B, 5A to 5B and 5C to 5D. 
